# Comparison of Fluid Replacement with Sterofundin ISO^®^ vs. Deltajonin^®^ in Infants Undergoing Craniofacial Surgery—A Retrospective Study

**DOI:** 10.3390/jcm12196404

**Published:** 2023-10-08

**Authors:** Benjamin Tan, Isabel Schütte, Michael Engel, Thomas Bruckner, Markus A. Weigand, Cornelius J. Busch

**Affiliations:** 1Department of Anesthesiology, Medical Faculty Heidelberg, Heidelberg University, 69120 Heidelberg, Germany; benjamin.tan@med.uni-heidelberg.de (B.T.); isabel.schuette@med.uni-heidelberg.de (I.S.); markus.weigand@med.uni-heidelberg.de (M.A.W.); 2Department of Oral and Maxillofacial Surgery, Medical Faculty Heidelberg, Heidelberg University, 69120 Heidelberg, Germany; michael.engel@med.uni-heidelberg.de; 3Institute of Medical Biometry and Informatics, Medical Faculty Heidelberg, Heidelberg University, 69120 Heidelberg, Germany

**Keywords:** craniectomy, isochloremic solution, hyperchloremia

## Abstract

In recent decades, infusion solutions such as NaCl 0.9% and lactate Ringer’s solution have been replaced in clinical practice. Since 2017, the national guidelines for perioperative infusion therapy in children recommend balanced isotonic solutions to maintain fluid balance. The composition of balanced infusion solutions varies with respect to their electrolyte content. Hyperchloremia may be mistaken for hypovolemia and may interfere with volume therapy in pediatric patients. Sterofundin ISO^®^ balanced solution contains 127 mmol/L chloride and may cause hyperchloremic acidosis if administered in large volumes. Objectives: The purpose of this study was to compare the effects of Sterofundin ISO^®^ (SF) therapy with the balanced isochloremic solution Deltajonin^®^ (DJ) (106 mmol/L chloride) on the acid–base status in infants undergoing craniofacial surgery. Methods: This retrospective, non-blinded study included 100 infants undergoing craniectomy due to isolated nonsyndromic sagittal craniosynostosis. The first 50 infants received Sterofundin ISO^®^. Due to changes in national guidelines, the infusion was changed to the isoionic Deltajonin^®^ in an additional 50 infants in 2017. Pre- and postoperative values of chloride, pH, base excess, bicarbonate, and albumin and phosphate were determined, and the strong-ion difference, strong-ion gap, anion gap, and weak acids were calculated. Results: Both groups were comparable in terms of their age, sex, underlying disease, preoperative electrolytes (except K at 3.9 ± 0.3 mmol/L (SF) vs. 4.1 ± 0.3 mmol/L (DJ) and lactate 8.7 ± 2.1 (SF) vs. 9.6 ± 2.6 mmol/L (DJ)). In the Sterofundin ISO^®^ group, hyperchloremic metabolic acidosis was observed in 19 patients, whereas only 2 infants in the Deltajonin^®^ group had hyperchloremic metabolic acidosis. The postoperative chloride level was 111 ± 2.7 mmol/L (SF) vs. 108 ± 2.4 mmol/L (DJ). The difference in anion gap was 12.5 ± 3.0 mmol/L (SF) vs. 14.6 ± 2.8 mmol/L (DJ), and the difference in SIDa (apparent strong-ion difference) was 30.9 mmol/L (SF) vs. 33.8 mmol/L (DJ). Conclusions: Hyperchloremic acidosis can be induced by the volume replacement with high-chloride-concentration crystalloids such as Sterofundin ISO^®^. This can be detected using the Stewart model.

## 1. Introduction

Craniofacial surgery in infants is often associated with significant intraoperative blood loss [1], which requires substantial amounts of intravenous fluid replacement that can alter ion concentrations in the blood and other body compartments. Because of the comparatively small compartment volumes, these patients are more susceptible to changes in their electrolyte metabolism. Therefore, the choice of fluid therapy may alter the electrolyte composition and acid–base status in these patients.

Side effects of hyperchloremia due to fluid therapy include increased rates of postoperative infection and renal dysfunction [2].

In adult patients undergoing major abdominal surgery, the use of a calcium-free balanced isochloremic crystalloid for fluid balance on the day of major surgery was associated with less postoperative morbidity, postoperative infections, renal failure requiring dialysis, blood transfusions, and electrolyte disturbances than in the 0.9% saline group [2]. Therefore, the type of crystalloid fluid replacement is a matter of debate.

In 2017, a revised version of the German national guidelines on perioperative infusion therapy was published, recommending balanced, isotonic, and isoionic solutions [3]. Therefore, in our institution, the crystalloid Sterofundin ISO^®^ (SF)(chloride 127.0 mmol/L, sodium 145.0 mmol/L, 24.0 mmol/L acetate, 5.0 mmol/L malate, corresponding to a theoretical osmolarity of 309 mosm/L) previously used was replaced by the more isoionic Deltajonin^®^ (DJ)(containing 108 mmol/L chloride, 140 mmol/L sodium, and 45 mmol/L lactate, corresponding to a theoretical osmolarity of 299 mosm/L).

To investigate whether the change in electrolyte solution caused a difference in the electrolyte composition and acid–base status in this population, we analyzed our hospital database and matched infants undergoing craniofacial surgery from two different time periods. One group received Sterofundin ISO^®^ prior to 2017 and the other group received Deltajonin^®^ after 2017 and the implementation of the new perioperative infusion therapy guidelines.

Our main hypothesis was that fluid resuscitation would lead to more hyperchloremia and hyperchloremic metabolic acidosis in infants with a balanced electrolyte solution with a chloride content greater than 120 mmol/L than a balanced electrolyte solution with a chloride content less than 120 mmol/L. 

Second, we wanted to better understand and describe the changes in the acid–base status induced by substitution of the various crystalloid solutions in conjunction with colloid substitution and blood transfusion.

Therefore, we chose the Stewart model of acid–base interpretation to further retrospectively analyze the collected data and describe the observed changes in the acid–base status.

Infants undergoing open strip craniectomy either receiving Sterofundin ISO^®^ oder Deltajonin^®^ were examined pre- and postoperatively. We report that hyperchloremic acidosis can be induced by volume replacement with high chloride concentration crystalloids such as Sterofundin ISO^®^.

## 2. Materials and Methods

Data were retrospectively obtained from the hospital database of Heidelberg University Hospital after approval by the local human-research ethics committee (S237-2009, German clinical trial registration number DRKS00028283). 

Inclusion criteria were all patients under the age of two suffering from isolated nonsyndromic sagittal craniosynostosis, receiving treatment with subtotal cranial vault remodeling in the time period between 2013 and 2019. Exclusion criteria were patients with syndromic craniosynostosis and patients older than two years. 

Patient demographics, pre-existing conditions, the duration of surgery, details of intraoperative fluids administered, and pre- and post-operative acid–base and electrolyte levels were recorded. Blood loss (Vloss) was estimated using the following formula: Vloss (mL) = BV × (Hbi − Hbe)/Hbi, (BV (blood volume) = 80 mL/kg body weight, Hbi = hemoglobin level before surgery, Hbe = lowest hemoglobin level during surgery) [4,5]. The composition of Sterofundin ISO^®^ and Deltajonin^®^ is given in Appendix A.

### 2.1. Anesthesia 

General anesthesia was induced with 0.3 µg/kg sufentanil, 5 mg/kg propofol, and 0.5 mg/kg atracurium and maintained with sevoflurane. Patients were ventilated with a FiO_2_ of 0.4, a tidal volume of 8 mL/kg, and a rate of 26–30/min to achieve normocapnia (35–40 mmHg) and normoxia. A second venous catheter and an arterial line (radial artery) were placed. Postoperative analgesia was achieved with Piritramid (initial dose 0.1 mg/kg body weight and 0.05 mg/kg for repetitive doses if needed).

### 2.2. Fluid Replacement

All patients received a balanced electrolyte solution containing 1% glucose (“ELEKTROLYT INFUSION SOLUTION 148 with Glucose 1 PED”, Serumwerk Bernburg AG, Bernburg, Germany; mmol/L: Na^+^ 140; K^+^ 4; Ca^2+^ 1; Mg^2+^ 1; Cl^−^ 118; acetate^−^ 30; glucose 55.5) as maintenance fluid during both time periods. Fluid intake was 4 mL/h per kg body weight, starting after the first access was established and continued until the end of surgery.

After induction of anesthesia, all patients received an initial bolus of 10 mL/kg body weight of fluid (Sterofundin ISO^®^ or Deltajonin^®^) to counteract fasting hypovolemia and anesthesia-induced hypotension. 

All further crystalloid fluid administration required was performed with the solutions studied, either Sterofundin ISO^®^ in 2013 to 2016 or Deltajonin^®^ in 2017 to 2019.

Additional fluid boluses of 10 mL/kg body weight were administered when signs of hypovolemia occurred (i.e., tachycardia (defined as an increase of >20% over the baseline heart rate of the previous 10 min), hypotension (defined as MAP < 40 mmHg), or urine flow < 1 mL/kg/h, BE < −5, and pH < 7.25), or a visible delta swing in arterial blood pressure was observed.

If more than 60 mL/kg BW of crystalloid fluid bolus was needed for hemodynamic stabilization, 20% human albumin was administered in the form of 5 mL/kg body weight bolus.

When Hb was <10 g/dL, red blood cells were administered with boluses of 5–10 mL/kg. Fresh frozen plasma (FFP) was administered in boluses of 5–10 mL/kg if estimated blood loss exceeded 30% of estimated blood volume or if the transfusion of red blood cells exceeded 40 mL/kg body weight. 

### 2.3. Catecholamines

The decision to administer additional catecholamines to further support a sudden drop in blood pressure was left to the attending anesthesiologist. When catecholamines were administered, this was either as a bolus of 1–6 μg norepinephrine or 1–2 mL of a 1:100 dilution of Akrinor^®^ (ratiopharm, Ulm, Germany).

### 2.4. Data Analysis

Blood gas analysis (RAPIDPoint 500, Siemens, Erlangen, Germany) and central laboratory results were recorded preoperatively and postoperatively.

Baseline albumin and phosphate values were obtained one day before surgery in the preoperative laboratory.

Baseline electrolyte, blood gas, acid–base status, lactate, and hemoglobin values were obtained at the first arterial-blood gas sampling after application of the arterial line before the start of surgery.

Postoperative albumin and phosphate levels were taken from the first postoperative laboratory, performed immediately after arrival in the pediatric intensive care unit or sometimes even directly in the operating room before transfer to the intensive care unit.

Post-interventional electrolyte, blood gas, acid–base status, lactate, and hemoglobin values were determined from the last arterial-blood gas collected in the operating room after surgical closure.

Parameters describing acid–base status using the Stewart model, such as the strong-ion difference, strong-ion gap, anion gap, and A- were calculated retrospectively from the collected data. Acidosis was defined as pH < 7.35 and a BE lower than <−2. Hyperchloremia was defined as >111 mmol/L and hyperchloremic acidosis as pH < 7.35, BE < −2, and chloride level > 111 mmol/L.

Because there are no established data to date on fluid replacement with Sterofundin ISO^®^ compared to Deltajonin^®^ in infants, the trial was planned as a pilot study without confirmative results. Therefore, the sample size was set at n = 50 per group to obtain a large enough sample to provide robust descriptive results.

Continuous data are expressed as mean ± standard deviation (SD), and for statistical analysis, a *t*-test was used to compare data between the two groups. Significance was assumed when *p* < 0.05. 

## 3. Results

The records of all 100 children who underwent linear craniectomy between 2013 and 2019 were used for data analysis. The mean age was 6.8 ± 3.7 months in the Sterofundin ISO^®^ compared (SF) group and 7.4 ± 2.4 months in the Deltajonin^®^ (DJ) group (Table 1). A total of 34 female and 66 male children were included, equally distributed between the two groups. There was no difference in blood loss between the two groups (245.6 ± 163.5 mL (SF) vs. 249.8 ± 151.7 mL (DJ)). The mean weight was lower in the SF group (7.8 ± 1.2 kg) than in the DJ group (8.4 ± 1.6 kg). In addition, the duration of the procedure was longer in the SF group (108.3 ± 29.2 min) than in the DJ group (81.4 ± 27.3 min). 

Taken together, the groups differed in terms of their weight and the duration of the procedure, but not in terms of age, sex distribution, or blood loss (Table 1; additional data are available in Appendix A).

Regarding baseline blood values, no differences were observed for the chloride, pH, base excess, or anion gap, as well as for SIDa (apparent strong-ion difference), SIDe (effective strong-ion difference), or SIG (difference of SIDa and SIDe; quantifies [unmeasured anions]–[unmeasured cations] of both strong and weak ions, albumin and A-). The groups differed only in their potassium and lactate levels without clinical relevance (Table 2).

Overall, the results show that both groups had comparable preoperative blood values (see Table 3). pH: potentia hydrogenii, SIDa: apparent strong-ion difference, SIDe: effective strong-ion difference, SIG: difference of SIDa and SIDe, A-: weak acid.

### 3.1. Fluid Replacement Data 

During surgery, all children received maintenance fluid (E148 G-1 Päd) with a mean volume of 14.22 (±8.95) mL/kg per hour in the SF group and 12.83 (±5.09) mL/kg per hour in the DJ group.

All children were given additional crystalloid fluid during surgery. On average, patients in the SF group received 42.0 ± 26.6 mL/kg body weight. The DJ group received an average of 48.2 ± 22.7 mL/kg body weight.

There were no differences between the two groups in the amount of fluid administered.

Forty-eight children in the SF group and fifty children in the DJ group received a transfusion of red blood cells. Twenty-nine children in the SF group and forty-four patients in the DJ group received albumin during the surgical procedure. FFP was administered to fourteen children in the SF group and seven children in the DJ group. In the SF group, seven patients received catecholamine support with norepinephrine or Akrinor^®^. In the DJ group, eleven patients received boli with Akrinor^®^ (Table 3). None of the patients in either group required further catecholamine support.

The detailed total amount of intraoperative fluid infused is shown in Table 3. 

### 3.2. Incidence of Hyperchloremia and Hyperchloremic Acidosis

Postoperative blood sample analysis data are shown in Table 4.

There was a postoperative difference in the chloride (SF: 111.0 ± 2.7 vs. DJ: 108.0 ± 2.4 mmol/L; Figure 1), anion gap (SF: 12.5 ± 3.0 vs. DJ: 14.6 ± 2. 8 mmol/L) and SIDa (SF: 30.9 ± 3.0 vs. DJ: 33.8 ± 2.0 mmol/L; Figure 2), as well as in the potassium levels (SF: 3.9 ± 0.4 vs. DJ: 4.2 ± 0.4 mmol/L). Furthermore, differences were observed in pCO_2_ (SF: 39.5 ± 4.7 vs. DJ: 41.9 ± 6.4 mmHg), albumin (SF: 34 ± 6.5 vs. DJ: 40.1 ± 6.4 g/L, Figure 3), A- (SF: 12.2 ± 1.7 vs. DJ: 14.2 ± 1.8 mmol/L), and SIDe (SF: 31.6 ± 3 vs. DJ: 34.5 ± 2.3 mmol/L). Hemoglobin was preoperatively lower in the DJ group (9.6 ± 1.0 g/dL vs. 10.8 ± 1.1 g/dL) but showed no difference postoperatively (SF: 11.4 ± 1.6 g/dL vs. DJ: 11.2 ± 1.4 g/dL; see also Table 5).

The figures for the values of chloride, pH, base excess, anion gap, SIDa, SIDe, SIG, albumin, and A- show the difference between the preoperative and postoperative electrolyte balance between the SF group and the DJ group (Figure 1, Figure 2 and Figure 3 and Appendix A).

A proportion of 23 of the 50 patients (46%) in the SF group showed postoperative hyperchloremia with chloride levels > 111 mmol/L. In the DJ group, 2 of 50 patients (4%) showed hyperchloremia with chloride levels > 111 mmol/L. In 19 patients in the SF group, hyperchloremic metabolic acidosis with chloride levels above 111 mmol/L and pH values < 7.35 was present postoperatively. In the DJ group, only two patients had hyperchloremic metabolic acidosis (see Table 4). It is also worth mentioning that 26 patients in the SF group had hypoalbuminemia < 37 g/L, compared with 17 in the DJ group.

These results show that patients receiving Sterofundin ISO^®^ as an additional fluid replacement were more likely to have hyperchloremia and hyperchloremic metabolic acidosis compared to Deltajonin^®^.

## 4. Discussion

This study investigated the effect of two different electrolyte solutions and their influence on plasmatic electrolyte concentrations and the acid–base status in infants during surgery. The main hypothesis was that the higher chloride content (>120 mmol/L) of Sterofundin ISO^®^ would lead to more hyperchloremia and metabolic acidosis compared to Deltajonin^®^, which has a chloride content of <120 mmol/L. The main finding of our study was that the administration of Sterofundin ISO^®^ leads to more cases of hyperchloremia but not to more metabolic acidosis.

### 4.1. Effects of Chloride Content of Electrolyte Solutions

Numerous studies have examined the effects of the chloride content of electrolyte solutions on plasmatic chloride concentrations in adults. In most of these studies, NaCl 0.9% was compared with balanced electrolyte solutions containing less than 120 mmol/L chloride. In most studies of surgical intensive care patients, a clear association between administration of 0.9% NaCl and hyperchloremia was found [6,7,8,9,10,11,12]. In contrast, the recent SOLAR study in adults undergoing colorectal or orthopedic surgery showed no differences between the administration of Lactate Ringer’s solution and NaCl 0.9% in hyperchloremia, postoperative renal failure, serious postoperative complications, or hospital mortality [13].

Hyperchloremia in adults and children can have several side effects, the most serious of which is acute kidney injury and the need for renal replacement therapy [2,8,9,14,15]. Because acute kidney injury correlates with increased mortality [16] and prolonged hospitalization, hyperchloremia or even the amount of chloride administered to a patient per se may be related to mortality, as has been pointed out [2,9,13,17].

Other side effects of hyperchloremia include hyperkalemia and metabolic acidosis [2,6,12], decreased renal blood flow [18,19], the increased release of inflammatory mediators in sepsis [20], increased need for catecholamine support [7], increased positive fluid balance [9], increased number of blood transfusions [2], prolonged need for mechanical ventilation and ICU stay [14], and increased health care costs [12]. 

Presumably because of the smaller difference in the chloride content compared with NaCl 0.9% in the aforementioned studies, the observed effects in our study were much smaller or negligible. Higher rates of hyperchloremia were observed in the Sterofundin ISO^®^ group, but there were no differences in the rates of metabolic acidosis or hyperkalemia, as described in the literature cited above comparing NaCl 0.9% with balanced electrolyte solutions.

In the present collective of healthy infants, no perioperative renal failure was observed. Because there was no change in metabolic acidosis, we doubt that the small difference in hyperchloremia would have had long-term effects on renal function. 

### 4.2. Comparison with Previous Studies

Comparison of the results of the present study with previously published data is difficult because most studies compared 0.9% NaCl solutions with balanced electrolyte solutions and, thus, compared electrolyte solutions with a much higher chloride difference. Weinberg et al. also compared two balanced electrolyte solutions (Hartmann’s solution vs. Plasmalyte-148^®^) in adult surgical patients undergoing major liver resection [21]. Both electrolyte solutions had a lower chloride content than the solutions compared in the present study (Hartmann’s solution 109 mmol/L, Plasmalyte-148^®^ 98 mmol/L). A higher chloride content was observed more frequently in the Hartmann group, but similar to our results, Weinberg et al. did not detect any effects on the acid–base status.

Specifically, the difference in chloride concentration in Weinberg’s study between the two groups was 106 vs. 108 mmol/L in the immediate postoperative period and 105 vs. 107 mmol/L after 24 h. The difference in chloride concentration was not significant. All these values are still within the normal range of plasmatic chloride concentration and are not hyperchloremic (>111 mmol/L) according to our definition. Therefore, it is not surprising that although there was a difference in chloride concentration between the two groups, it may have been too small to affect the acid–base status and lead to hyperchloremic acidosis.

Both crystalloid solutions in the present study contained more chloride compared with the solutions used in the study of Weinberg et al. Accordingly, the measured postoperative chloride values were higher in the present study. The values of the present study are also still considered to be within the normal range, but toward the upper limit of normochloremia than the observed values of Weinberg et al. The absolute difference between the two groups compared in the present study and that of Weinberg was similar, approximately 2–3 mmol/L, which may explain the lack of effect on the acid–base status from a clinical point of view. The comparison of the low-chloride group in the study of Weinberg with the “high content” group in the present study could show differences in the acid–base status.

In summary, both studies demonstrated that even small differences in the chloride content of balanced electrolyte solutions affect plasmatic chloride concentrations in adult and pediatric surgical patients. Whether these differences in chloride concentration have clinical implications cannot be answered from the available data. 

### 4.3. Acid–Base Effects and the Stewart Approach

To better understand the effects of the different solutions on the acid–base status in this study, the Stewart approach was used. In both groups, the SIG value was approximately −0.6, which excludes the effect of unmeasured anions on the acid–base status.

A lower SIDa value was observed in the SF group than in the DJ group. Accordingly, the SIDa value decreased more in the SF group than in the DJ group. This reflects the higher increase in chloride concentration from the preoperative to postoperative values in the SF group and the absolutely higher chloride concentration in postoperative values between the two groups. 

Interestingly, albumin showed a greater decrease (∆ of −10 vs. −4 g/L) and an absolutely lower postoperative value (34 vs. 40 g/L) in the SF group compared with the DJ group. Consistent with these results, SIDe decreased more in the SF group from preoperative to postoperative values and had lower absolute values compared with the DJ group. Because albumin acts as a weak acid, according to the Stewart approach, the observed hypoalbuminemia in the SF group would lead to metabolic alkalosis, which would compensate for the hyperchloremic acidosis and could lead overall to the unobserved differences in pH and base excess between the two groups. 

These results suggest that Sterofundin ISO^®^ administration resulted in hyperchloremic acidosis that was compensated by hypoalbuminous alkalosis. No such effect was observed in the DJ group. 

The difference in the chloride content of the solutions studied explains the differences in the measured chloride concentrations and SIDa values, but otherwise the compositions of the two solutions do not explain the differences in the decrease in albumin and SIDe. Because both solutions were albumin-free and were administered in the same amount, the decrease in albumin and SIDe should be similar in both groups, making the effect of hyperchloremic acidosis more apparent. But as demonstrated, this was not the case.

The smaller decrease in albumin and SIDe in the DJ group cannot be explained by albumin substitution, because both groups received the same amounts of human albumin or fresh frozen plasma. Another explanation could be a lower blood loss in the DJ group and a correspondingly lower albumin loss. Since blood loss was not accurately measured during surgery, this remains speculative. In contrast, the estimated blood loss and transfused units of red blood cells did not differ between the two groups. 

The operation was shorter in the DJ group, possibly resulting in less substitution by crystalloids and less dilution of the plasmatic albumin concentration. This hypothesis is not supported by the data presented, because there were no differences between the two groups in the total volume administered per hour and body weight.

Because all procedures were performed by the same surgeon over the years, one could speculate that he improved his surgical technique over time, as reflected in the shorter operative times, and that this probably enabled him to operate more efficiently and with less blood loss. Otherwise, from the data we collected, there is no obvious explanation for the different drop and postoperative levels of albumin and SIDe in the two groups. Preoperatively, there was no difference in these values.

The collection and analysis of independent variables of the Stewart approach revealed the effects of hyperchloremia on the acid–base status and uncovered unexplained hypoalbuminemia that compensated for and masked the hyperchloremic acidosis in the SF group.

### 4.4. Relevance of Hyperchloremia Caused by Sterofundin ISO

From a practical point of view, the question remains whether the higher chloride content of Sterofundin ISO^®^ and the associated hyperchloremia have clinical implications. 

Disma et al. compared two intraoperative fluid regimens in 229 children aged 1 to 36 months undergoing elective moderate to major surgery: Sterofundin^®^ (127 mmol/L chloride) with 1% glucose solution and NaCl 0.9% solution with 1% glucose solution [22]. Postoperatively, lower chloride levels and more physiological pH and base excesses were observed in the Sterofundin^®^ group. In addition to these expected results for Sterofundin^®^, Disma et al. identified an age of less than 10 months and the infusion of large volumes greater than 46.7 mL/kg body weight of Sterofundin^®^ as possible risk factors for the development of hyperchloremia. This volume of fluid administration could easily be achieved, for example, in a septic child, in whom current guidelines recommend that up to 40–60 mL/kg be administered as a fluid bolus in the first hour [23], making the chloride content of Sterofundin^®^ more clinically relevant in this patient population and the risk of hyperchloremia more worrisome, especially since the risk of acute kidney injury is high in patients with sepsis [24]. The results of the present study are consistent with those of the study of Disma et al., in which almost 50% of patients receiving Sterofundin ISO^®^ in amounts greater than 42 mL/kg body weight developed hyperchloremia, whereas the same amount of Deltajonin^®^ resulted in hyperchloremia in only 4% of patients.

Overall, these results indicate that Sterofundin ISO^®^ causes hyperchloremia, albeit to a lesser extent than NaCl 0.9%. When administered in large quantities, the chloride content of Sterofundin ISO^®^ is still likely to be too high and could be a risk factor for electrolyte disturbances, especially in infants with sepsis. 

### 4.5. Limitations

This retrospective study has several limitations in that it is susceptible to selection, misclassification, and information bias. Because of the surgical setting, the accurate quantification of blood loss was not possible. Because the blood gas analyzers in the operating room did not measure albumin and phosphate concentrations, these values had to be determined in the core laboratory. There was a difference in the maintenance fluid infused that could affect postoperative electrolyte concentrations.

## 5. Conclusions

In a population of infants undergoing linear craniectomy, resuscitation with two different balanced crystalloid solutions that had a high and normal chloride content had effects on the postoperative chloride content and acid–base status. Resuscitation with a solution with a higher chloride content resulted in higher postoperative plasmatic chloride concentrations and lower SIDa. The use of a balanced electrolyte solution with an isoionic chloride content was less likely to result in hyperchloremia and promoted a more stable acid–base balance. Therefore, we continued to use Deltajonin^®^ as the standard crystalloid infusion for these patients.

The effects of different balanced electrolyte solutions with varying chloride contents and their effects on short- and long-term morbidity and mortality should be studied more extensively in critically ill infants.

## Figures and Tables

**Figure 1 jcm-12-06404-f001:**
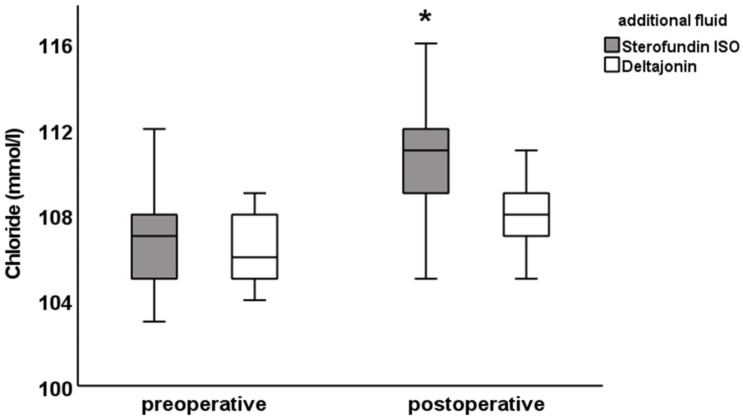
Chloride pre- and postoperative values. Blood samples were measured using an arterial blood gas analyzer right before and at the end of surgery. Preoperative chloride values were similar, while postoperative chloride was higher after Sterofundin ISO when compared to corresponding Deltajonin. Data are presented as mean ± SD; * *p* < 0.05 vs. Deltajonin.

**Figure 2 jcm-12-06404-f002:**
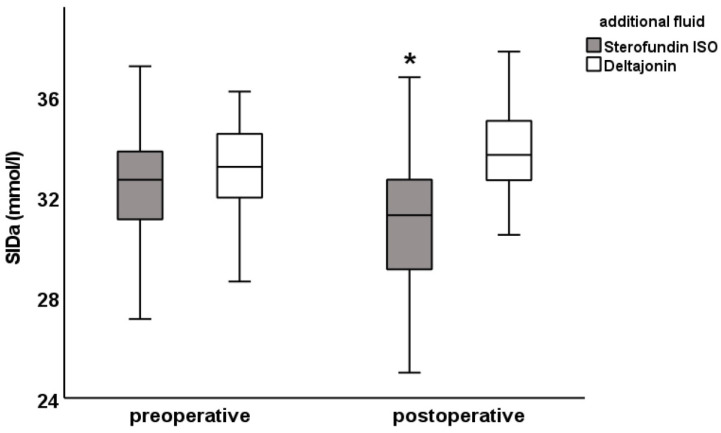
SIDa pre- and postoperative values: SIDa was calculated with [SIDa] = [Na^+^] + [K^+^] − [Cl^−^] − [Lac^−^]. Preoperative SIDa values were similar, while there was postoperatively a difference between Sterofundin ISO and corresponding Deltajonin; data are presented as mean ± SD; * *p* < 0.05 vs. Deltajonin.

**Figure 3 jcm-12-06404-f003:**
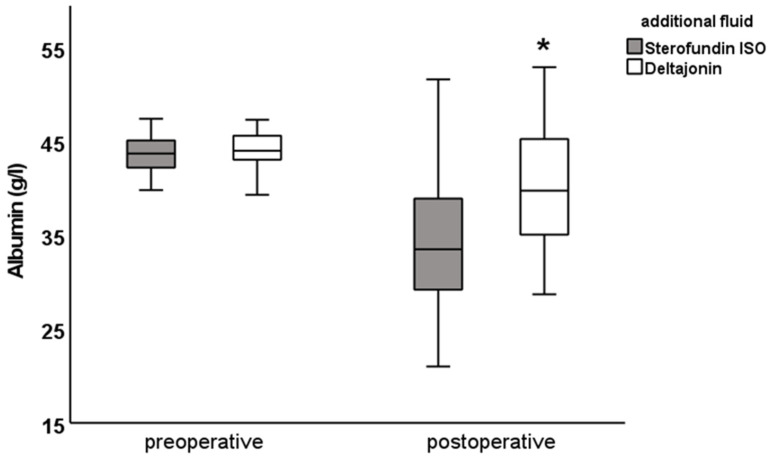
Albumin values pre- and postoperative. Blood samples were taken one day before and right after surgery. Preoperative albumin values were similar. Postoperatively, there was a difference between the Sterofundin ISO and Deltajonin group; data are presented as mean ± SD; * *p* < 0.05 vs. Sterofundin ISO.

**Table 1 jcm-12-06404-t001:** Characteristics of the study cohort.

	SF (n = 50)	DJ (n = 50)	*p*-Value
Age (month)	6.3 ± 3.7	7.4 ± 2.4	0.0886
Sex (m/f)	33/17	33/17	1
Weight (kg)	7.8 ± 1.2	8.4 ± 1.6	0.025
Duration surgery (min)	108.3 ± 29.2	86.3 ± 31.2	<0.001
Blood loss (mL)	245.6 ± 163.5	249.8 ± 151.7	0.447

SF: Sterofundin ISO^®^, DJ: Deltajonin^®^.

**Table 2 jcm-12-06404-t002:** Baseline blood values.

	SF (n = 50)	DJ (n = 50)	*p*-Value
Chloride [mmol/L]	106.9 ± 2.2	106.3 ± 1.6	0.055
pH	7.35 ± 0.06	7.35 ± 0.07	0.445
Base excess	−4.2 ± 1.9	−4.0 ± 1.7	0.250
Anion gap [mmol/L]	12.4 ± 2.6	12.9 ± 2.2	0.187
SIDa [mmol/L]	32.5 ± 2.5	33.1 ± 1.8	0.062
SIDe [mmol/L]	36.2 ± 1.1	36.8 ± 1.6	0.124
SIG [mmol/L]	−3. 9 ± 2.8	−3.6 ± 2.2	0.348
Albumin [g/L]	43.4 ± 2	44.1 ± 2	0.115
A- [mmol/L]	15.1 ± 0.7	15.3 ± 0.6	0.170
Potassium [mmol/L]	3.9 ± 0.3	4.1 ± 0.3	0.031
Lactate [mg/dL]	8.7 ± 2.1	9.6 ± 2.6	0.033

**Table 3 jcm-12-06404-t003:** Intraoperative fluid administration and catecholamine support.

	SF (n = 50)	DJ (n = 50)	*p*-Value
Additional fluid (mL)	332.1 ± 225.8	389.3 ± 156.0	0.07
Additional fluid (mL/kg)	42.0 ± 26.6	48.2 ± 22.7	0.11
Basal fluid (E148 G-1 Päd)(mL)	187.8 ± 102	145.4 ± 63.7	0.007
Basal fluid (E148 G-1 Päd) (mL/kg h)	14.2 ± 9	12.8 ± 5.1	0.951
	**SF (n = 29)**	**DJ (n = 44)**	
Albumin (mL) Albumin (mL/kg)	97.8 ± 62.7 12.2 ± 6.8	89.3 ± 48.1 11.4 ± 6.7	0.615 0.308
	**SF (n = 48)**	**DJ (n = 50)**	
PRBC (mL) PRBC (mL/kg)	233.9 ± 112.8 29.7 ± 12.5	267.3 ± 117.1 33 ± 16	0.077 0.133
	**SF (n = 14)**	**DJ (n = 7)**	
FFP (mL) FFP (mL/kg)	183.6 ± 121.3 21.8 ± 13.5	248.6 ± 55.5 30 ± 3.9	0.054 0.025
	**SF (n = 3)**	**DJ (n = 11)**	
Akrinor (mg)	0.12 ± 0.51	0.36 ± 0.23	0.10
	**SF (n = 4)**	**DJ (n = 0)**	
Noradrenaline [µg]	5.2 ± 3.11	-	

PRBC: packed red blood cell, FFP: fresh frozen plasma.

**Table 4 jcm-12-06404-t004:** Differences from pre-to postoperative values.

	SF (n = 50)	DJ (n = 50)	*p*-Value
Chloride (mmol/L)	+3.9 ± 2.4	+2 ± 2.8	<0.001
pH	−0.04 ± 0.08	−0.05 ± 0.1	0.414
Bicarbonate (mmol/L)	+1.5 ± 2.9	+1.1 ± 2.3	0.191
Base Excess (mmol/L)	−2.1 ± 3.1	−1.7 ± 2.4	0.228
Anion Gap	+0.09 ± 2.1	+1.7 ± 2.3	<0.001
SIDa (mmol/L)	−1.6 ± 2.6	+0.6 ± 2.6	<0.001
	**SF (n = 25)**	**DJ (n = 28)**	***p*-value**
Albumin (g/L)	−9.5 ± 7	−4 ± 7	0.003
A^-^ (mmol/L)	−2.9 ± 1.9	−1.1 ± 2	<0.001
SIDe (mmol/L)	−5.3 ± 3.3	−2.3 ± 2.8	<0.001
SIG	+3.5 ± 2.7	+2.8 ± 2.1	0.159

**Table 5 jcm-12-06404-t005:** Hemoglobin in g/dL pre- and postoperative.

	SF (n = 50)	DJ (n = 50)
Hb preoperative (g/dL)	10.8 ± 1.1	9.6 ± 1.0
Hb postoperative (g/dL)	11.4 ± 1.6	11.2 ± 1.4
*p*-value pre- vs. postoperative	<0.001	<0.001

## Data Availability

The datasets used and analyzed during the current study are available from the corresponding author on request.

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
