# Peer review of "Comparison of Fluid Replacement with Sterofundin ISO® vs. Deltajonin® in Infants Undergoing Craniofacial Surgery—A Retrospective Study"

_jcm, 2023, doi:10.3390/jcm12196404_

Round 1

Reviewer 1 Report

This article is interesting, but here are some comments.

1. In the abstract and materials and methods sections, the research design is not clearly explained. It is recommended to refine these sections, particularly in terms of describing the research design, specifying patient age, outlining inclusion and exclusion criteria, and so on.

2. In line 160-168, although there is no significant difference, it would be better if this section were also displayed in table form.

3. In lines 188-189, the administration of catecholamine support is not included in the existing table. As a suggestion, this information can be added into Table 2, with an additional titled ‘administration of Catecholamine Support’ alongside the original title ‘intraoperative fluid administration.

4. In lines 159, 229, 316, 341, The frequent use of the phrase 'in conclusion' throughout the article is excessive. Please include this word only at the end of the discussion.

Reviewer 2 Report

Thank you for permitting me to  review this manuscript 

In this manuscript the authors compared  retrospectively two regimens of fluid therapy in craniofacial surgery for children  and found that crystalloids such as sterofundin , might induce hyperchloremic acidosis 

Line 301 , please describe SIG (may be I missed it before)

Here are my suggestions 

Line 48 please precise which country ? I guess Germany? 

PLease escribe and elaborate earlier in the introduction the hyperchloremic state and its negative impact 

Line 115 , why the target of hb transfusion in 10 ? please explain 

as this target is lower elsewhere 

Please elaborate the sample size calculation simply setting 50 is not enough

Since there was difference in the two groups within the cohort a matching  with propension score would have been useful  please explain why no matching was used 

figure 1 and  2 the asterisk is bettter to be for sterofundin

Round 2

Reviewer 2 Report

The authors have responded to my queries 
